# Do Deep Nets Really Need to be Deep?

**Lei Jimmy Ba**
University of Toronto
jimmy@psi.utoronto.ca

**Rich Caruana**
Microsoft Research
rcaruana@microsoft.com

## Abstract

Currently, deep neural networks are the state of the art on problems such as speech recognition and computer vision. In this paper we empirically demonstrate that shallow feed-forward nets can learn the complex functions previously learned by deep nets and achieve accuracies previously only achievable with deep models. Moreover, in some cases the shallow nets can learn these deep functions using the same number of parameters as the original deep models. On the TIMIT phoneme recognition and CIFAR-10 image recognition tasks, shallow nets can be trained that perform similarly to complex, well-engineered, deeper convolutional models.

## 1 Introduction

You are given a training set with 1M labeled points. When you train a shallow neural net with one fully connected feed-forward hidden layer on this data you obtain 86% accuracy on test data. When you train a deeper neural net as in [1] consisting of a convolutional layer, pooling layer, and three fully connected feed-forward layers on the same data you obtain 91% accuracy on the same test set.

What is the source of this improvement? Is the 5% increase in accuracy of the deep net over the shallow net because: a) the deep net has more parameters; b) the deep net can learn more complex functions given the same number of parameters; c) the deep net has better inductive bias and thus learns more interesting/useful functions (e.g., because the deep net is *deeper* it learns hierarchical representations [5]); d) nets without convolution can't easily learn what nets with convolution can learn; e) current learning algorithms and regularization methods work better with deep architectures than shallow architectures[8]; f) all or some of the above; g) none of the above?

There have been attempts to answer this question. It has been shown that deep nets coupled with unsupervised layer-by-layer pre-training [10] [19] work well. In [8], the authors show that depth combined with pre-training provides a good prior for model weights, thus improving generalization. There is well-known early theoretical work on the representational capacity of neural nets. For example, it was proved that a network with a large enough single hidden layer of sigmoid units can approximate any decision boundary [4]. Empirical work, however, shows that it is difficult to train shallow nets to be as accurate as deep nets. For vision tasks, a recent study on deep convolutional nets suggests that deeper models are preferred under a parameter budget [7]. In [5], the authors trained shallow nets on SIFT features to classify a large-scale ImageNet dataset and found that it was difficult to train large, high-accuracy, shallow nets. And in [17], the authors show that deeper models are more accurate than shallow models in speech acoustic modeling.

In this paper we provide empirical evidence that shallow nets are capable of learning the same function as deep nets, and in some cases with the same number of parameters as the deep nets. We do this by first training a state-of-the-art deep model, and then training a shallow model to mimic the deep model. The mimic model is trained using the model compression method described in the next section. Remarkably, with model compression we are able to train shallow nets to be as accurate as some deep models, even though we are not able to train these shallow nets to be as accurate as the deep nets when the shallow nets are trained directly on the original labeled training data. If a shallow net *with the same number of parameters* as a deep net can learn to mimic a deep net with high fidelity, then it is clear that the function learned by that deep net does not really have to be deep.

## 2 Training Shallow Nets to Mimic Deep Nets

### 2.1 Model Compression

The main idea behind model compression [3] is to train a compact model to approximate the function learned by a larger, more complex model. For example, in [3], a single neural net of modest size could be trained to mimic a much larger ensemble of models—although the *small* neural nets contained 1000 times fewer parameters, often they were just as accurate as the ensembles they were trained to mimic. Model compression works by passing unlabeled data through the large, accurate model to collect the scores produced by that model. This synthetically labeled data is then used to train the smaller mimic model. The mimic model is not trained on the original labels—it is trained to learn the *function* that was learned by the larger model. If the compressed model learns to mimic the large model perfectly it makes exactly the same predictions and mistakes as the complex model.

Surprisingly, often it is not (yet) possible to train a small neural net on the original training data to be as accurate as the complex model, *nor as accurate as the mimic model*. Compression demonstrates that a small neural net could, *in principle*, learn the more accurate function, but current learning algorithms are unable to train a model with that accuracy from the original training data; instead, we must train the complex intermediate model first and then train the neural net to mimic it. Clearly, when it is possible to mimic the function learned by a complex model with a small net, the function learned by the complex model wasn't truly too complex to be learned by a small net. This suggests to us that the complexity of a learned model, and the size and architecture of the representation best used to learn that model, are different things.

### 2.2 Mimic Learning via Regressing Logits with L2 Loss

On both TIMIT and CIFAR-10 we use model compression to train shallow mimic nets using data labeled by either a deep net, or an ensemble of deep nets, trained on the original TIMIT or CIFAR-10 training data. The deep models are trained in the usual way using softmax output and cross-entropy cost function. The shallow mimic models, however, instead of being trained with cross-entropy on the 183 $p$ values where $p_k = e^{z_k} / \sum_j e^{z_j}$ output by the softmax layer from the deep model, are trained directly on the 183 log probability values $z$, also called logits, *before* the softmax activation.

Training on logits, which are logarithms of predicted probabilities, makes learning easier for the student model by placing equal emphasis on the relationships learned by the teacher model *across all of the targets*. For example, if the teacher predicts three targets with probability $[2 \times 10^{-9}, 4 \times 10^{-5}, 0.9999]$ and those probabilities are used as prediction targets and cross entropy is minimized, the student will focus on the third target and tend to ignore the first and second targets. A student, however, trained on the logits for these targets, $[10, 20, 30]$, will better learn to mimic the detailed behaviour of the teacher model. Moreover, consider a second training case where the teacher predicts logits $[-10, 0, 10]$. After softmax, these logits yield the same predicted probabilities as $[10, 20, 30]$, yet clearly the teacher models the two cases very differently. By training the student model directly on the logits, the student is better able to learn the internal model learned by the teacher, without suffering from the information loss that occurs from passing through logits to probability space.

We formulate the SNN-MIMIC learning objective function as a regression problem given training data $\{(x^{(1)}, z^{(1)}),...,(x^{(T)}, z^{(T)})\}$:

$$\mathcal{L}(W, \beta) = \frac{1}{2T} \sum_t ||g(x^{(t)}; W, \beta) - z^{(t)}||_2^2, \tag{1}$$

where $W$ is the weight matrix between input features $x$ and hidden layer, $\beta$ is the weights from hidden to output units, $g(x^{(t)}; W, \beta) = \beta f(W x^{(t)})$ is the model prediction on the $t^{th}$ training data point and $f(\cdot)$ is the non-linear activation of the hidden units. The parameters $W$ and $\beta$ are updated using standard error back-propagation algorithm and stochastic gradient descent with momentum.

We have also experimented with other mimic loss functions, such as minimizing the KL divergence $\mathrm{KL}(p_{\text{teacher}} \| p_{\text{student}})$ cost function and L2 loss on probabilities. Regression on logits outperforms all the other loss functions and is one of the key techniques for obtaining the results in the rest of this

paper. We found that normalizing the logits from the teacher model by subtracting the mean and dividing the standard deviation of each target across the training set can improve L2 loss slightly during training, but normalization is not crucial for obtaining good student mimic models.

## 2.3 Speeding-up Mimic Learning by Introducing a Linear Layer

To match the number of parameters in a deep net, a shallow net has to have more non-linear hidden units in a single layer to produce a large weight matrix $W$. When training a large shallow neural network with many hidden units, we find it is very slow to learn the large number of parameters in the weight matrix between input and hidden layers of size $O(HD)$, where $D$ is input feature dimension and $H$ is the number of hidden units. Because there are many highly correlated parameters in this large weight matrix, gradient descent converges slowly. We also notice that during learning, shallow nets spend most of the computation in the costly matrix multiplication of the input data vectors and large weight matrix. The shallow nets eventually learn accurate mimic functions, but training to convergence is very slow (multiple weeks) even with a GPU.

We found that introducing a bottleneck linear layer with $k$ *linear* hidden units between the input and the non-linear hidden layer sped up learning dramatically: we can factorize the weight matrix $W \in \mathbb{R}^{H \times D}$ into the product of two low-rank matrices, $U \in \mathbb{R}^{H \times k}$ and $V \in \mathbb{R}^{k \times D}$, where $k << D, H$. The new cost function can be written as:

$$\mathcal{L}(U, V, \beta) = \frac{1}{2T} \sum_t ||\beta f(UVx^{(t)}) - z^{(t)}||_2^2 \tag{2}$$

The weights $U$ and $V$ can be learnt by back-propagating through the linear layer. This re-parameterization of weight matrix $W$ not only increases the convergence rate of the shallow mimic nets, but also reduces memory space from $O(HD)$ to $O(k(H + D))$.

Factorizing weight matrices has been previously explored in [16] and [20]. While these prior works focus on using matrix factorization in the last output layer, our method is applied between the input and hidden layer to improve the convergence speed during training.

The reduced memory usage enables us to train large shallow models that were previously infeasible due to excessive memory usage. Note that the linear bottleneck can only reduce the representational power of the network, and it can always be absorbed into a single weight matrix $W$.

# 3 TIMIT Phoneme Recognition

The TIMIT speech corpus has 462 speakers in the training set, a separate development set for cross-validation that includes 50 speakers, and a final test set with 24 speakers. The raw waveform audio data were pre-processed using 25ms Hamming window shifting by 10ms to extract Fourier-transform-based filter-banks with 40 coefficients (plus energy) distributed on a mel-scale, together with their first and second temporal derivatives. We included +/- 7 nearby frames to formulate the final 1845 dimension input vector. The data input features were normalized by subtracting the mean and dividing by the standard deviation on each dimension. All 61 phoneme labels are represented in tri-state, i.e., three states for each of the 61 phonemes, yielding target label vectors with 183 dimensions for training. At decoding time these are mapped to 39 classes as in [13] for scoring.

## 3.1 Deep Learning on TIMIT

Deep learning was first successfully applied to speech recognition in [14]. Following their framework, we train two deep models on TIMIT, DNN and CNN. DNN is a deep neural net consisting of three fully connected feedforward hidden layers consisting of 2000 rectified linear units (ReLU) [15] per layer. CNN is a deep neural net consisting of a convolutional layer and max-pooling layer followed by three hidden layers containing 2000 ReLU units [2]. The CNN was trained using the same convolutional architecture as in [6]. We also formed an ensemble of nine CNN models, ECNN.

The accuracy of DNN, CNN, and ECNN on the final test set are shown in Table 1. The error rate of the convolutional deep net (CNN) is about 2.1% better than the deep net (DNN). The table also shows the accuracy of shallow neural nets with 8000, 50,000, and 400,000 hidden units (SNN-8k,

SNN-50k, and SNN-400k) trained on the original training data. Despite having up to 10X as many parameters as DNN, CNN, and ECNN, the shallow models are 1.4% to 2% less accurate than the DNN, 3.5% to 4.1% less accurate than the CNN, and 4.5% to 5.1% less accurate than the ECNN.

## 3.2 Learning to Mimic an Ensemble of Deep Convolutional TIMIT Models

The most accurate *single* model that we trained on TIMIT is the deep convolutional architecture in [6]. Because we have no unlabeled data from the TIMIT distribution, we use the same 1.1M points in the train set as unlabeled data for compression by throwing away the labels.[1] Re-using the 1.1M train set reduces the accuracy of the student mimic models, increasing the gap between the teacher and mimic models on test data: model compression works best when the unlabeled set is very large, and when the unlabeled samples do not fall on train points where the teacher model is likely to have overfit. To reduce the impact of the gap caused by performing compression with the original train set, we train the student model to mimic a more accurate ensemble of deep convolutional models.

We are able to train a more accurate model on TIMIT by forming an ensemble of nine deep, convolutional neural nets, each trained with somewhat different train sets, and with architectures of different kernel sizes in the convolutional layers. We used this very accurate model, ECNN, as the teacher model to label the data used to train the shallow mimic nets. As described in Section 2.2 the logits (log probability of the predicted values) from each CNN in the ECNN model are averaged and the average logits are used as final regression targets to train the mimic SNNs.

We trained shallow mimic nets with 8k (SNN-MIMIC-8k) and 400k (SNN-MIMIC-400k) hidden units on the re-labeled 1.1M training points. As described in Section 2.3, to speed up learning both mimic models have 250 linear units between the input and non-linear hidden layer—preliminary experiments suggest that for TIMIT there is little benefit from using more than 250 linear units.

## 3.3 Compression Results For TIMIT

|  | Architecture | # Param. | # Hidden units | PER |
|---|---|---|---|---|
| SNN-8k | 8k + dropout<br>trained on original data | ~12M | ~8k | 23.1% |
| SNN-50k | 50k + dropout<br>trained on original data | ~100M | ~50k | 23.0% |
| SNN-400k | 250L-400k + dropout<br>trained on original data | ~180M | ~400k | 23.6% |
| DNN | 2k-2k-2k + dropout<br>trained on original data | ~12M | ~6k | 21.9% |
| CNN | c-p-2k-2k-2k + dropout<br>trained on original data | ~13M | ~10k | **19.5%** |
| ECNN | ensemble of 9 CNNs | ~125M | ~90k | **18.5%** |
| SNN-MIMIC-8k | 250L-8k<br>no convolution or pooling layers | ~12M | ~8k | **21.6%** |
| SNN-MIMIC-400k | 250L-400k<br>no convolution or pooling layers | ~180M | ~400k | **20.0%** |

Table 1: Comparison of shallow and deep models: phone error rate (PER) on TIMIT core test set.

The bottom of Table 1 shows the accuracy of shallow mimic nets with 8000 ReLUs and 400,000 ReLUs (SNN-MIMIC-8k and -400k) trained with model compression to mimic the ECNN. Surprisingly, shallow nets are able to perform as well as their deep counterparts when trained with model compression to mimic a more accurate model. A neural net with one hidden layer (SNN-MIMIC-8k) can be trained to perform as well as a DNN with a similar number of parameters. Furthermore, if we increase the number of hidden units in the shallow net from 8k to 400k (the largest we could

train), we see that a neural net with one hidden layer (SNN-MIMIC-400k) can be trained to perform comparably to a CNN, even though the SNN-MIMIC-400k net has no convolutional or pooling layers. This is interesting because it suggests that a large single hidden layer without a topology custom designed for the problem is able to reach the performance of a deep convolutional neural net that was carefully engineered with prior structure and weight-sharing *without any increase in the number of training examples*, even though the same architecture trained on the original data could not.

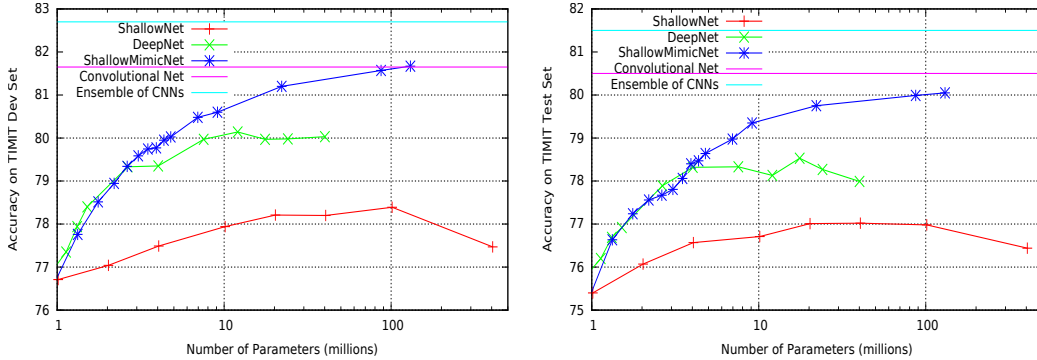

Figure 1: Accuracy of SNNs, DNNs, and Mimic SNNs vs. # of parameters on TIMIT Dev (left) and Test (right) sets. Accuracy of the CNN and target ECNN are shown as horizontal lines for reference.

Figure 1 shows the accuracy of shallow nets and deep nets trained on the original TIMIT 1.1M data, and shallow mimic nets trained on the ECNN targets, as a function of the number of parameters in the models. The accuracy of the CNN and the teacher ECNN are shown as horizontal lines at the top of the figures. When the number of parameters is small (about 1 million), the SNN, DNN, and SNN-MIMIC models all have similar accuracy. As the size of the hidden layers increases and the number of parameters increases, the accuracy of a shallow model trained on the original data begins to lag behind. The accuracy of the shallow mimic model, however, matches the accuracy of the DNN until about 4 million parameters, when the DNN begins to fall behind the mimic. The DNN asymptotes at around 10M parameters, while the shallow mimic continues to increase in accuracy. Eventually the mimic asymptotes at around 100M parameters to an accuracy comparable to that of the CNN. The shallow mimic never achieves the accuracy of the ECNN it is trying to mimic (because there is not enough unlabeled data), but it is able to match or exceed the accuracy of deep nets (DNNs) *having the same number of parameters* trained on the original data.

## 4 Object Recognition: CIFAR-10

To verify that the results on TIMIT generalize to other learning problems and task domains, we ran similar experiments on the CIFAR-10 Object Recognition Task[12]. CIFAR-10 consists of a set of natural images from 10 different object classes: airplane, automobile, bird, cat, deer, dog, frog, horse, ship, truck. The dataset is a labeled subset of the 80 million tiny images dataset[18] and is divided into 50,000 train and 10,000 test images. Each image is 32x32 pixels in 3 color channels, yielding input vectors with 3072 dimensions. We prepared the data by subtracting the mean and dividing the standard deviation of each image vector to perform global contrast normalization. We then applied ZCA whitening to the normalized images. This pre-processing is the same used in [9].

### 4.1 Learning to Mimic an Ensemble of Deep Convolutional CIFAR-10 Models

We follow the same approach as with TIMIT: An ensemble of deep CNN models is used to label CIFAR-10 images for model compression. The logit predictions from this teacher model are used as regression targets to train a mimic shallow neural net (SNN). CIFAR-10 images have a higher dimension than TIMIT (3072 vs. 1845), but the size of the CIFAR-10 training set is only 50,000 compared to 1.1 million examples for TIMIT. Fortunately, unlike TIMIT, in CIFAR-10 we have access to unlabeled data from a similar distribution by using the superset of CIFAR-10: the 80 million tiny images dataset. We add the first one million images from the 80 million set to the original 50,000 CIFAR-10 training images to create a 1.05M mimic training (transfer) set.

| | Architecture | # Param. | # Hidden units | Err. |
|---|---|---|---|---|
| DNN | 2000-2000 + dropout | ~10M | 4k | 57.8% |
| SNN-30k | 128c-p-1200L-30k + dropout input&hidden | ~70M | ~190k | 21.8% |
| single-layer feature extraction | 4000c-p followed by SVM | ~125M | ~3.7B | 18.4% |
| CNN[11] (no augmentation) | 64c-p-64c-p-64c-p-16lc + dropout on lc | ~10k | ~110k | 15.6% |
| CNN[21] (no augmentation) | 64c-p-64c-p-128c-p-fc + dropout on fc and stochastic pooling | ~56k | ~120k | 15.13% |
| teacher CNN (no augmentation) | 128c-p-128c-p-128c-p-1kfc + dropout on fc and stochastic pooling | ~35k | ~210k | **12.0%** |
| ECNN (no augmentation) | ensemble of 4 CNNs | ~140k | ~840k | **11.0%** |
| SNN-CNN-MIMIC-30k trained on a single CNN | 64c-p-1200L-30k with no regularization | ~54M | ~110k | **15.4%** |
| SNN-CNN-MIMIC-30k trained on a single CNN | 128c-p-1200L-30k with no regularization | ~70M | ~190k | **15.1%** |
| SNN-ECNN-MIMIC-30k trained on ensemble | 128c-p-1200L-30k with no regularization | ~70M | ~190k | **14.2%** |

Table 2: Comparison of shallow and deep models: classification error rate on CIFAR-10. Key: c, convolution layer; p, pooling layer; lc, locally connected layer; fc, fully connected layer

CIFAR-10 images are raw pixels for objects viewed from many different angles and positions, whereas TIMIT features are human-designed filter-bank features. In preliminary experiments we observed that non-convolutional nets do not perform well on CIFAR-10, no matter what their depth. Instead of raw pixels, the authors in [5] trained their shallow models on the SIFT features. Similarly, [7] used a base convolution and pooling layer to study different deep architectures. We follow the approach in [7] to allow our shallow models to benefit from convolution while keeping the models as shallow as possible, and introduce a single layer of convolution and pooling in our shallow mimic models to act as a feature extractor to create invariance to small translations in the pixel domain. The SNN-MIMIC models for CIFAR-10 thus consist of a convolution and max pooling layer followed by fully connected 1200 linear units and 30k non-linear units. As before, the linear units are there only to speed learning; they do not increase the model's representational power and can be absorbed into the weights in the non-linear layer after learning.

Results on CIFAR-10 are consistent with those from TIMIT. Table 2 shows results for the shallow mimic models, and for much deeper convolutional nets. The shallow mimic net trained to mimic the teacher CNN (SNN-CNN-MIMIC-30k) achieves accuracy comparable to CNNs with multiple convolutional and pooling layers. And by training the shallow model to mimic the ensemble of CNNs (SNN-ECNN-MIMIC-30k), accuracy is improved an additional 0.9%. The mimic models are able to achieve accuracies previously unseen on CIFAR-10 with models with so few layers. Although the deep convolutional nets have more hidden units than the shallow mimic models, because of weight sharing, the deeper nets with multiple convolution layers have fewer parameters than the shallow fully connected mimic models. Still, it is surprising to see how accurate the shallow mimic models are, and that their performance continues to improve as the performance of the teacher model improves (see further discussion of this in Section 5.2).

# 5 Discussion

## 5.1 Why Mimic Models Can Be More Accurate than Training on Original Labels

It may be surprising that models trained on targets predicted by other models can be more accurate than models trained on the original labels. There are a variety of reasons why this can happen:

- If some labels have errors, the teacher model may eliminate some of these errors (i.e., censor the data), thus making learning easier for the student.

- Similarly, if there are complex regions in $p(y|X)$ that are difficult to learn given the features and sample density, the teacher may provide simpler, soft labels to the student. Complexity can be *washed away by filtering* targets through the teacher model.

- Learning from the original hard 0/1 labels can be more difficult than learning from a teacher's conditional probabilities: on TIMIT only one of 183 outputs is non-zero on each training case, but the mimic model sees non-zero targets for most outputs on most training cases, and the teacher can spread uncertainty over multiple outputs for difficult cases. The uncertainty from the teacher model is more informative to the student model than the original 0/1 labels. This benefit is further enhanced by training on logits.

- The original targets may depend in part on features not available as inputs for learning, but the student model sees targets that depend only on the input features; the targets from the teacher model are a function *only* of the available inputs; the dependence on unavailable features has been eliminated by filtering targets through the teacher model.

The mechanisms above can be seen as forms of regularization that help prevent overfitting in the student model. Typically, shallow models trained on the original targets are more prone to overfitting than deep models—they begin to overfit before learning the accurate functions learned by deeper models even with dropout (see Figure 2). If we had more effective regularization methods for shallow models, some of the performance gap between shallow and deep models might disappear. Model compression appears to be a form of regularization that is effective at reducing this gap.

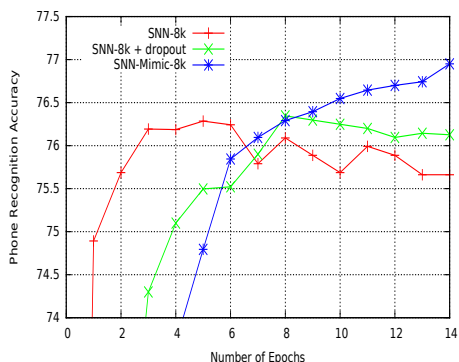

Figure 2: Shallow mimic tends not to overfit.

## 5.2 The Capacity and Representational Power of Shallow Models

Figure 3 shows results of an experiment with TIMIT where we trained shallow mimic models of two sizes (SNN-MIMIC-8k and SNN-MIMIC-160k) on teacher models of different accuracies. The two shallow mimic models are trained on the same number of data points. The only difference between them is the size of the hidden layer. The x-axis shows the accuracy of the teacher model, and the y-axis is the accuracy of the mimic models. Lines parallel to the diagonal suggest that increases in the accuracy of the teacher models yield similar increases in the accuracy of the mimic models. Although the data does not fall perfectly on a diagonal, there is strong evidence that the accuracy of the mimic models continues to increase as the accuracy of the teacher model improves, suggesting that the mimic models are not (yet) running out of capacity. When training on the same tar-

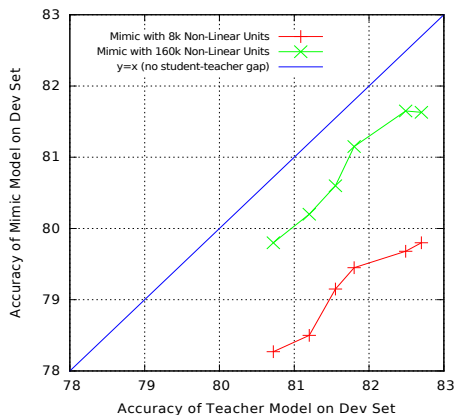

Figure 3: Accuracy of student models continues to improve as accuracy of teacher models improves.

gets, SNN-MIMIC-8k always perform worse than SNN-MIMIC-160K that has 10 times more parameters. Although there is a consistent performance gap between the two models due to the difference in size, the smaller shallow model was eventually able to achieve a performance comparable to the larger shallow net by learning from a better teacher, and the accuracy of both models continues to increase as teacher accuracy increases. This suggests that shallow models with a number of parameters comparable to deep models probably are capable of learning even more accurate functions

if a more accurate teacher and/or more unlabeled data become available. Similarly, on CIFAR-10 we saw that increasing the accuracy of the teacher model by forming an ensemble of deep CNNs yielded commensurate increase in the accuracy of the student model. We see little evidence that shallow models have limited capacity or representational power. Instead, the main limitation appears to be the learning and regularization procedures used to train the shallow models.

### 5.3 Parallel Distributed Processing vs. Deep Sequential Processing

Our results show that shallow nets can be competitive with deep models on speech and vision tasks. In our experiments the deep models usually required 8–12 hours to train on Nvidia GTX 580 GPUs to reach the state-of-the-art performance on TIMIT and CIFAR-10 datasets. Interestingly, although some of the shallow mimic models have more parameters than the deep models, the shallow models train much faster and reach similar accuracies in only 1–2 hours.

Also, given parallel computational resources, at run-time shallow models can finish computation in 2 or 3 cycles for a given input, whereas a deep architecture has to make sequential inference through each of its layers, expending a number of cycles proportional to the depth of the model. This benefit can be important in on-line inference settings where data parallelization is not as easy to achieve as it is in the batch inference setting. For real-time applications such as surveillance or real-time speech translation, a model that responds in fewer cycles can be beneficial.

## 6   Future Work

The tiny images dataset contains 80 millions images. We are currently investigating whether, if by labeling these 80M images with a teacher, it is possible to train shallow models with no convolutional or pooling layers to mimic deep convolutional models.

This paper focused on training the shallowest-possible models to mimic deep models in order to better understand the importance of model depth in learning. As suggested in Section 5.3, there are practical applications of this work as well: student models of small-to-medium size and depth can be trained to mimic very large, high-accuracy deep models, and ensembles of deep models, thus yielding better accuracy with reduced runtime cost than is currently achievable without model compression. This approach allows one to adjust flexibly the trade-off between accuracy and computational cost.

In this paper we are able to demonstrate empirically that shallow models can, *at least in principle*, learn more accurate functions without a large increase in the number of parameters. The algorithm we use to do this—training the shallow model to mimic a more accurate deep model, however, is awkward. It depends on the availability of either a large unlabeled dataset (to reduce the gap between teacher and mimic model) or a teacher model of very high accuracy, or both. Developing algorithms to train shallow models of high accuracy directly from the original data without going through the intermediate teacher model would, if possible, be a significant contribution.

## 7   Conclusions

We demonstrate empirically that shallow neural nets can be trained to achieve performances previously achievable only by deep models on the TIMIT phoneme recognition and CIFAR-10 image recognition tasks. Single-layer fully connected feedforward nets trained to mimic deep models can perform similarly to well-engineered complex deep convolutional architectures. The results suggest that the strength of deep learning may arise in part from a good match between deep architectures and current training procedures, and that it may be possible to devise better learning algorithms to train more accurate shallow feed-forward nets. For a given number of parameters, depth may make learning easier, but may not always be essential.

**Acknowledgements** We thank Li Deng for generous help with TIMIT, Li Deng and Ossama Abdel-Hamid for the code for their deep convolutional TIMIT model, Chris Burges, Li Deng, Ran Gilad-Bachrach, Tapas Kanungo and John Platt for discussion that significantly improved this work, David Johnson for help with the speech model, and Mike Aultman for help with the GPU cluster.

## Footnotes

[1]That SNNs can be trained to be as accurate as DNNs using *only* the original training data highlights that it *should* be possible to train accurate SNNs on the original training data given better learning algorithms.

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
