[Reviews · NeurIPS 2014]

Submitted by Assigned_Reviewer_15

The authors show empirical results on TIMIT and CIFAR-10 that shallow nets trained to mimic the outputs of DNNs and CNNs achieve comparable accuracy on these tasks. The paper is clearly written and makes compelling arguments. The contribution is significant because it suggests that SNNs are capable of learning complex functions that were thought to be learnable only with DNNs or CNNs. This means that better training algorithms have yet to be devised for SNNs. Some minor typos/comments:
- line 85: a either -> either
- line 125: [16] also uses the factorization to speed up training and reduce memory, the difference with your work is only output layer versus input layer.
- line 323: reasons reasons -> reasons
Summary: Good paper, clearly written, of significant importance to the deep learning community.

Submitted by Assigned_Reviewer_17

The question that the authors pose is essentially why do Neural Net work
and if it is necessary to use a large number of hidden layers. More precisely
the paper attempts to answer the question if a large number of hidden layers
gets better results because current parameter estimation techniques (SGD and
pre-training) favors deep Neural Nets or a large number of hidden layers is
simply better for other reasons (hierarchical feature extraction,..).
To answer this question the authors first train regular deep Neural Networks
(both MLP's and CNN's) and then train a shallow Neural Net to mimic the deep
network. This is done by using the log outputs of the deep network as targets
for the shallow network. The output layer of the shallow network does not use
a (nonlinear) activation function and the loss function is l2.
The experiments are done on two tasks: TIMIT (small scale phone recognition task)
and CIFAR-10 (image recognition). The result shows that shallow nets trained
to mimic deep nets are significantly better than a shallow net trained the
conventional way and getting close to the original deep model.

Pros:
The question that the authors trying to answer is relevant for understanding
Neural Networks and their parameter estimation. It is interesting idea to
apply model compression [3] to obtain a shallow network.

Cons:
The data sets used in these experiments are rather small. I'm not 100% sure
that the results would be the same for large training set. It could be very
well that deep Neural Networks profit more from large training sets and that
miming deep network with shallow networks will not work as well as for small
data sets.

The paper by Vinyals and Morgan is related to this work. But also there,
only small data sets were used.

Other References:
Vinyals, Oriol, and Nelson Morgan.
"Deep vs. wide: depth on a budget for robust speech recognition." INTERSPEECH. 2013.
Summary: The question that the authors trying to answer is relevant for understanding
Neural Networks and their parameter estimation. The data sets used in these
experiments are rather small. I'm not 100% sure that the results would be the
same for large training set. It could be very well that deep Neural Networks
profit more from large training sets and that miming deep network with shallow
networks will not work as well as for small data sets.

Submitted by Assigned_Reviewer_29

This paper presents experiments with shallow network trained to mimic
the predictions of deeper, more complex networks trained on the same
data. When trained to predict the pre-softmax outputs of the
"teacher" networks, these mimic networks are able to approximate the
performance of the deeper networks, albeit with more parameters. They
consistently outperform networks with the same architecture trained
the cross-entropy objective function.

The paper presents an interesting -- and perhaps useful -- set of
experiments, but the theoretical analysis is extremely weak. There
are a number of bold claims that in my opinion are not demonstrated by
the experimental work, and many speculative statements, the truth of
which there has been little effort to investigate. The paper is
reasonably original, though methods similar to the mimic-network have
been presented before.

The paper is clearly written and reasonably well-organised, although
the colloquial style ("what is the source of this magic?") in the
introduction is not to my taste. The strong part of the paper is the
demonstration that a shallower neural network can achieve good
performance when trained to mimic a deeper "teacher" network, or
ensemble. This is good news for software engineers tasked with
building low-footprint systems, although, as noted, related techniques
have already been employed [20]. It is presented as a surprising
finding, but I am not sure why this should be so: not dissimilar
methods have also had success, such as the KL-HMM. As in that model,
it may be seen that the teacher network is providing a large amount of
information to the learner in the vector z(t), as in the KL-HMM where
the p(t) are used directly, so it is somewhat misleading to say that
the learner is simply trained to mimic the (one-best) predictions of
the teacher network - it's being given much more information than
that. Analagously, the purported benefit of the KL-HMM is that the
posteriors generated by one NN are helpful targets for another (abeit
much simpler) model. (Note, I am aware that the KL-HMM differs
materially from the mimic-model).

Section 5.1 is particularly weak in my opinion. If I were to write it
with my own theories as to why the mimic models work, they would be
quite different to those being presented, but no more speculative.
The first theory, that the teacher eliminates errors, could easily be
tested, but isn't. The second, "the complexity in the data set has
been washed away" makes no sense at all -- what does this mean? The
third seems more plausible, but again, this could easily be tested.
"Model compression appears to be a form of regularization" needs much
more support, as it underpins the core claim that if only shallow
models could be regularized more successfully, then they could be
trained to perform as well as the deep models without the help of a
teacher.

However, this is a diversion from my main point, which is that I
believe that the theoretical premise of the paper is flawed. As
widely known and cited [4], a neural network with a single hidden
layer can approximate any decision boundary, with a large enough
number of hidden units. This point, which seems to be the core
experimental finding of the paper, is not novel. However, the "large
enough" number of hidden units implies the requirement for an
abritrarily large amount of training data. Isn't one of the key
strengths of a deep network not the deep functional architecture per
se, but rather its ability to make more efficient use of the available
data when learning the weights, due to its hierarchical structure?
This would certainly be the conclusion I would be tempted to draw from
Table 1, for example, where you demonstrate that a deep network
considerably outperforms a shallow network with the same number of
parameters, but that the deep structure isn't actually necessary to
produce equivalently-good decision boundaries. So my answer to the
question posed in the second paragraph of the introduction would
probably be (g). There argument that the shallow network could
perform better, if only the regularization were more effective or the
learning algorithm improve, just isn't supported by the experiments.
And sadly there is no theoretical justification either.

This is a shame, because as I said, the experiements are nice, and
there are lots of future directions the work could go. In speech
recognition, for example, it would be interesting to investigate
whether the learner/teacher methodology would work across different
training sets: for example, for speaker adaptation; or multi-lingual
or cross-domain ASR. There are other interesting questions that are
left unanswered: for example, what would happen if you had a much
larger quantity of unlabelled data, and a teacher network trained on a
labelled subset. Would the mimic network assymptotically approach the
performance of the teacher, or given the extra data, could the method
exceed the performance of the teacher? I suspect not, but it would be
good to find out.

In summary, this paper an interesting concept, but it needs to be
presented quite differently, with more theoretical analysis and less
speculation.

Some specific points:

090 -> "training on these logit values makes learning easier for the
shallow net by placing emphasis on confident predictions" - really?

308 -> "shallow net achieves accuracy comparable to CNNs" - this is a bit misleading as the accuracy is some
way off that of the teacher CNN here.

Summary: This is an interesting paper with some useful experiments, but the analysis is often speculative, and I do not believe that many of the major claims are justified by the experiments.

Submitted by Assigned_Reviewer_42

Excellent paper, very clear and easy to understand.
It is also a very timely paper with the kind of studies that should be more often presented.
Actually, there are even many more parameters people can play with to further improve the performance of Shallow Neural Networks (SNN)!
empirical comparisons are made across many different neural net topologies, also paying with the number of parameters, on a quite reasonable size data set.
In the future though, I believe we should forget that old/obsolete notion of "keeping the number of parameters constant" as long as we guarantee that all methods being compared actually work at their "optimal" point.
Summary: Very good and clear paper, also very informative.
To help the "younger reader", I would suggest to:

1) Briefly recall what a "Shallow Neural Network" (quite an old term, used a long time ago, I believe) is, compared to a "Multilayer Perceptron".
2) Insist of bit more on reference [4], which is not very well know and still very key to the present paper, and summarize conclusions of that paper.

Submitted by Assigned_Reviewer_43

I thought this paper was really interesting. You tried to get at a very important question using ingenious experiments.
There are a few mistakes (IMO) and several places where you could improve readability. Be particularly careful not to generalize too much from your results: the fact that you failed to train some network does not mean it cannot be done, for instance.

Here are some comments, most important first.

Errors:
1) At the end of 2.3 you say: "Because the liner layer can always be absorbed into the the weight matrix W, even after adding the linear layer the model has similar representational power as the original shallow net before adding the linear layer." Not true! The linear bottleneck reduces the representational power. Perhaps what you wanted to say is: "The linear bottle neck can only reduce the representational power of the network, and it can always be absorbed into a single weight matrix W."
2) In 5.1 and 5.2 the references to the figures (2 and 3) are wrong in my copy, appearing as 5.1 and 5.2.

Omissions:
1) In 3.1 we really need to know a little more about the CNNs used on TIMIT. Were they convolutional in frequency, time or both?
2) In 3.1 we need to know more about the ensemble of nine CNNs. How did they differ? By structure or by training data?
3) In 3.2, how did you make a single target logit vector from the nine networks?

I have worked hard to improve the paper. Here are my suggestions.

Suggested Improvements (in order of appearance in the paper):
1. Introduction
The introduction starts very abruptly. I suggest starting "Imagine you are given ...".
Second para uses "magic" unnecessarily. I suggest "improvement".
Right at the end of the Introduction you have a crucial statement, which is the main motivation for the paper: "If a shallow net with the same number of parameters as a deep net can learn to mimic a deep net with high fidelity, then it is clear that the function learned by the deep net does not really have to be deep." I worry that this could be read as a statement about deep nets not being necessary in general, and I suggest changing one word so that the sentence ends "... the function learned by that deep net does not really have to be deep" to make it clear that it would be a property of the specific function, and not deep nets in general.

2.1
the scores predicted by the model -> the scores produced by the model
(The scores may be predictions of the labels, but they are produced by the model.)

2.2
"labeled by _a_ either a deep net"

2.2
I would really like to know whether the target values produced by the teacher network include the offset corresponding to the normalization of the SoftMax outputs.

3.3
Two uses of "comparably to" which would be better as "as well as".

3.3
"... even though the same architecture trained on the original data can not"
That is too strong a claim. I suggest "could not", making a summary of your experiences.

Figure1 (and the other figures)
You really ought to improve the look of the figures. The text should be twice the size, the lines should be much thicker.

PER and Accuracy
Table 1 is about phone error rates, Figure 1 is "Accuracy". I think I know what PER means, but you do not define accuracy. I tried (100-PER) but that does not reliably translate the relevant points from Table 1 to Figure 1. Explain.

5.1
"there are mislabeled frames introduced by the HMM forced-alignment procedure"
Could you make it clear what is the status of this? eg
We suspect that there are mislabeled frames introduced by the HMM forced-alignment procedure
We believe that there are mislabeled frames introduced by the HMM forced-alignment procedure
It is well known that there are mislabeled frames introduced by the HMM forced-alignment procedure

5.1 (a comment)
"learning from the original hard 0/1 labels can be more difficult than learning from the teacher’s conditional probabilities"
I wonder whether some of the benefit can be gained by replacing the hard labels with simply softened versions? say 0.95 instead of 1, and 0.05/182 for the rest? Or use the teacher's value instead of 1, and distribute the rest equally??

5.2 Figure 3
Why not draw the diagonal? It fits quite well.

5.2
"This suggests that shallow models with a number of parameters comparable to deep models are likely capable of learning even more accurate functions ..."
Rather than implying that it ought to suggest the same to the reader, I propose
"This suggests to us that ..."

6 Future Work
"Medium size and depth student models can be trained _to mimic_ to mimic very large, high accuracy deep models."
I suggest (_emphasis_ added only to show the changes)
Student models with medium size and depth _could_ be trained to mimic very large, high accuracy deep models.

6
"Under such scheme, one is able to adjust the trade-off of performance and computational cost on the existing models."
I suggest
"Under such _a_ scheme, one is able to adjust the trade-off of accuracy for computational cost."

7 Conclusions
'a good "impedance" match'
I do not think that the work "impedance" helps most readers here.
Summary: A thought-provoking paper with some rough edges.
Should be published and discussed.
Author Feedback
Author rebuttal: We thank all of the reviewers for their comments. We are gratified that several reviewers think the work is thought provoking and different from the typical NIPS submission.

We thank reviewers 15, 42, and 43 for their suggestions of how to improve the paper and will implement those accordingly. We thank reviewer 17 for suggesting the reference to the work of Vinyals and Morgan 2013; we will add that citation. As several reviewers suggest, we will modify the discussion to better connect our work to prior work. We will also increase the emphasis on the universal approximation theorem from 1990 as suggested by reviewer 42.

Reviewer 17 gave the lowest score (5/10). Their main concern seems to be that because the two data sets we use (TIMIT and CIFAR-10) are “small” the results may not generalize to larger datasets. We also are not 100% sure the results will be the same with larger training sets, and can see how the train-set-size argument could go either way. As the train set size gets very large, deeper models may learn additional detail that shallow models might have more difficulty representing or learning. On the other hand, as train set size increases, learning may asymptote, giving shallow models an opportunity to “catch up”. Larger train sets would reduce the variance of the deep teacher models (i.e., reduce overfitting), effectively acting as a regularizer that promotes smoothness, and this may make it easier for the student model to mimic the function of the deep model. Although larger training sets may allow larger deep models with more free parameters to be trained, the number of free parameters allowed in the shallow mimic models would also increase. Finally, if the size of the transfer set increases with train set size, the gap between the deep teacher model and the shallow student model might also be reduced. Reviewer 17 also suggests that the work is unlikely to have significant impact despite being technically sound. We respectfully disagree. Although the question of deep vs. shallow learning is not new, we believe the approach followed in this paper is not incremental. The main goal of this work is to better understand why deep learning works, and we think this is a very important topic for continued progress in the field.

We appreciate the effort that went into reviewer 43’s many suggestions of how to improve the papers readability, and completely agree with this reviewer that we should be careful not to generalize too much from the results. We will modify the paper to reflect this. We also agree there are rough edges in the paper and are eager to improve these. We will reword the end of 2.3 to be more clear about the impact of adding a linear layer to representational power. We will also provide more detail describing the teacher ensemble experiments (e.g., the ensemble teacher model was made by averaging the normalized logits from ensemble members). As mentioned in comment 5.1, we did try the soft target experiments and the results were much worse than using the soft label from teacher model.

Once again, thanks for the reviews.

… We were just about to upload the response above when we noticed that a 5th(!) review had been added later during the response period. Here is a brief response to that review: Reviewer 29 says the “theoretical analysis is very weak”. In line 51 we state that the paper presents an empirical demonstration that shallow models can learn the functions learned by deep models using a similar number of parameters as those deep models. We do not believe this kind of result can (yet) be proved theoretically for real data sets? We agree with the Reviewer 29’s comments (and the comments of other reviewers) that we should be careful not to overstate claims and that the language and tone of the paper can be improved. We also agree there are many interesting follow-up experiments suggested by this work, some of which are currently underway. However, doing these other experiments correctly is more complex than Reviewer 29 suggests, and it would not be possible to incorporate the additional methodology and results into a single NIPS-length paper. Like the other reviews, we believe the current paper and results are strong enough to stand on their own. Although we discuss model compression as a regularizer in Section 5.1 and present empirical results that show model compression reduces overfitting in shallow models (Figure 1), we agree with Reviewer 29 that much more work, both empirical and theoretical, needs to be done in this area. And we agree there is similarity between model compression and the training procedure in KL-HMM, though of course the goal and results are very different. We are happy to add references to this work and discuss similarities between the methods.